# Time Convolutional Network-Based Maneuvering Target Tracking with Azimuth–Doppler Measurement

**DOI:** 10.3390/s24010263

**Published:** 2024-01-02

**Authors:** Jianjun Huang, Haoqiang Hu, Li Kang

**Affiliations:** School of Intelligent Information Processing, Shenzhen University, Shenzhen 518060, China; huangjj_atr@sina.com.cn (J.H.); 2100432050@email.szu.edu.cn (H.H.)

**Keywords:** azimuth and Doppler, convolutional neural network, deep learning algorithm, maneuvering targets tracking

## Abstract

In the field of maneuvering target tracking, the combined observations of azimuth and Doppler may cause weak observation or non-observation in the application of traditional target-tracking algorithms. Additionally, traditional target tracking algorithms require pre-defined multiple mathematical models to accurately capture the complex motion states of targets, while model mismatch and unavoidable measurement noise lead to significant errors in target state prediction. To address those above challenges, in recent years, the target tracking algorithms based on neural networks, such as recurrent neural networks (RNNs), long short-term memory (LSTM) networks, and transformer architectures, have been widely used for their unique advantages to achieve accurate predictions. To better model the nonlinear relationship between the observation time series and the target state time series, as well as the contextual relationship among time series points, we present a deep learning algorithm called recursive downsample–convolve–interact neural network (RDCINN) based on convolutional neural network (CNN) that downsamples time series into subsequences and extracts multi-resolution features to enable the modeling of complex relationships between time series, which overcomes the shortcomings of traditional target tracking algorithms in using observation information inefficiently due to weak observation or non-observation. The experimental results show that our algorithm outperforms other existing algorithms in the scenario of strong maneuvering target tracking with the combined observations of azimuth and Doppler.

## 1. Introduction

With the increasing application of radar maneuvering target-tracking technology in both civil and military domains, how to use the data measured by radar sensors to obtain more accurate target state estimation has become a prominent research focus. Furthermore, the single-station passive positioning technology applied in target tracking utilizes the radar sensor that passively receives the source radiation signal to enable the positioning of the radiation source and is highly regarded for its strong concealment and broad application [1]. In the past, pure azimuth passive positioning technology was commonly used for target localization and tracking [2]. However, relying solely on the nonlinear relationship between angle measurements and target states made it challenging to obtain accurate target state estimation. Subsequently, researchers discovered that introducing new observables such as angle change rate and Doppler frequency could enhance localization accuracy [3,4]. For instance, combining azimuth and Doppler observables enables more precise target state estimation.

However, using the combined observations of azimuth and Doppler has certain requirements on the relative positions of the observation point and the target trajectory. In specific cases, the observation point will experience weak observation or non-observation during some relative motion periods [5,6], as shown in Figure 1. When a maneuvering target moves with a uniform or uniformly variable speed along the line connecting it to the observation point, both the azimuth and Doppler measurements have a rate of change of 0. During these periods, the target becomes unobservable as there is no significant change in the measured information. Similarly, when the maneuvering target follows a uniform circular motion relative to the observation point, the Doppler measurement remains constant, and only the azimuth measurement provides some observable information, this observation is considered weak due to the lack of Doppler information, which limits the accuracy of target state estimation during such periods.

In short, the single-station passive positioning technique based on the combination of azimuth and Doppler measurement has some limitations in the observation range of the maneuvering target tracking. Meanwhile, the maneuvering target states are always uncertain and complex in practicality, and the single-model traditional algorithms like extended Kalman filter (EKF) and unscented Kalman filter (UKF) applied in the field of weak maneuvering target tracking are difficult to achieve the accurate estimation of such complex maneuvering target states [7]. To solve this kind of complex strong maneuvering target tracking problem, multiple models (MM) algorithms and their various variants have been proposed and widely used [8]. Taking the interactive multiple-model algorithm (IMM) as an example [9,10], it usually needs to predefine a variety of target maneuvering models, model probabilities, and model transfer probabilities for the estimation of target states. However, in the specific application of this traditional target tracking algorithm, there is the problem of delayed model estimation when the target maneuvering state changes abruptly [11], coupled with the fact that the multi-model algorithm essentially needs to set up the motion models in advance just as the single-model target tracking algorithm does, and the inaccuracy of the preset motion model may also occur in the face of complex maneuvering situations, as well as the existence of observation noise in the radar sensors themselves, and various other adverse factors have a negative impact on this interactive multi-model algorithm [12]. Moreover, particle filter algorithms are extensively employed in tracking applications. For instance, the paper [13] proposes a method called a cost-referenced particle filter, which is utilized to estimate the state of discrete dynamic stochastic systems by dynamically optimizing user-defined cost functions, and it also presents a novel particle selection algorithm suitable for parallel computing, addressing the primary limitations of particle filter resampling technology. Additionally, a study [14] proposes a shadow filter that departs from the statistical basis of Kalman filtering, instead relying on deterministic dynamics to address tracking issues, and it delves into an analysis of the proposed filter method’s performance concerning its parameter influence as well.

In recent years, with the advancement of deep learning, the research on utilizing neural network modeling algorithms to break through the limitations of traditional target tracking algorithms has significantly expanded [15]. Leveraging the distinctive advantages of neural networks [16], it becomes feasible to dispense with the requirement of predefining the motion models beforehand and accomplish end-to-end predictions between observations and maneuvering target states. A previous study with a maneuvering trajectory prediction method that employed a backpropagation neural network (BPNN) was introduced to combine the historical trajectory of the target to capture the target’s motion patterns and generate predicted trajectories [17]. Furthermore, the articles [18,19,20] focus on addressing the challenges that arise from the inherent uncertainty in both maneuvering target states and the measurement information faced by traditional target tracking algorithms and presenting methodologies that better model the long-term dependence among sequence data through the gating mechanism. Subsequently, the articles [21,22] address the limitation of the long short-term memory (LSTM) model in capturing the global nature of the target maneuvering state by proposing the use of the transformer architecture which captures both long-term and short-term dependence of the target state, further enhancing the accuracy of target tracking algorithms.

To achieve an accurate estimation of strong maneuvering target states based on the combined observations of azimuth and Doppler, the first step is to use a time series of observations in the target motion state prediction to address the challenge of insufficient time information obtained from the azimuth–Doppler information at a single moment. The prediction of time series typically involves three fundamental structures, namely Recurrent Neural Networks (RNN), Transformer-based Networks (TBN), and Temporal Convolutional Networks (TCN) [23,24,25]. For leveraging the distinctive property where temporal relationships are largely preserved even after downsampling a time series into two subsequences, we propose a recursive downsample convolution interactive learning neural network (RDCINN) based on the Convolutional Neural Network (CNN) architecture [26] designed to address the challenge of motion states estimation. Our approach involves several key operations to extract motion features from the input observation time series. We first apply a full-connection layer and a position-coding layer to perform temporal coding operations on the input. Then, we proceed with recursive downsampling, temporal convolution, and interactive learning operations. In each layer, multiple convolutional filters are employed to extract motion features from the downsampling time series. By combining these rich features gathered from multiple resolutions, we can effectively address the issue of weak observation or non-observation encountered in traditional maneuvering target tracking algorithms based on the combined observations of azimuth and Doppler. Finally, the utilization of a binary tree structure in our model contributes to an increased depth of temporal feature extraction which allows for effective modeling of the nonlinear mapping relationship between high-noise observation time series and complex maneuvering states. 

## 2. Problem Formulation

As in previous studies on maneuvering target tracking using deep learning approaches [19,20,21], our simulation scenarios are set on a 2D plane. In this setup, the radar observation point passively receives azimuth and Doppler velocity, and is positioned at the origin O. We assumed that $s_{k}$ and $z_{k}$ are the k-th momentary maneuvering target state vector and observation vector, respectively.

$\left(x_{k},y_{k}\right)$ denotes the position of a k-th moment maneuvering target in the X-Y plane, $\left({\dot{x}}_{k},\;{\dot{\;y}}_{k}\right)$ represents the corresponding velocity. And ${{[\theta }_{k},\;{\;d}_{k}]}^{T}$ represents the azimuth and Doppler velocity of the measurements $z_{k}$ and is expressed as [27]:(1)$$\left[\begin{array}{c} \theta_{k}\\ d_{k} \end{array}\right]=\left[\begin{array}{c} \arctan\left(\frac{y_{k}}{x_{k}}\right)\\ {\dot{x}}_{k}{\mathrm{cos\theta}}_{k}+{\dot{y}}_{k}sin\theta_{k} \end{array}\right]+\left[\begin{array}{c} n_{\theta ,k}\\ n_{d,k} \end{array}\right]$$
where $n_{\theta ,k}\sim N\left(0,\sigma_{\theta }^{2}\right)$, $n_{d,k}\sim N\left(0,\sigma_{d}^{2}\right)$, $\sigma_{\theta }$, $\sigma_{d}$ are the standard deviations of the Gaussian noise of the azimuth and Doppler measurement, respectively.

To perform target tracking using a deep learning approach, it is essential to generate an extensive dataset of trajectories and observations for training the network model [28]. This dataset enables the modeling of the nonlinear relationship between target states and observation information, ultimately facilitating the accurate estimation of maneuvering target states. Typically, the dataset is generated based on the state equations and observation equations as follows:(2)$$\left\{\begin{array}{c} s_{k}=Fs_{k-1}+n_{s,k}\\ z_{k}=h(s_{k})+n_{z,k} \end{array}\right.$$
where $n_{s,k}$, $n_{z,k}$ denote state transfer noise and observation noise and are expressed as follows, respectively:(3)$$n_{z,k}={\left[n_{\theta ,k},n_{d,k}\right]}^{T}$$
(4)$$n_{s,k}={\left[n_{pos,k},n_{pos,k},n_{vel,k},n_{vel,k}\right]}^{T}$$
where $n_{pos,k}=\frac{T^{2}}{2}\alpha_{k}$, $n_{vel,k}=T\alpha_{k}$, T is the radar sensor sampling interval time. $\alpha_{k}\sim N(0,\sigma_{s}^{2})$ denotes the maneuvering acceleration noise, which follows a Gaussian distribution. In the state equation $s_{k}=Fs_{k-1}+n_{s,k}$, we have incorporated two maneuver models, namely the constant velocity (CV) motion and the constant turning (CT) motion. The definitions of these models are as follows:(5)$$F_{CV}=\left[\begin{array}{cc} \begin{array}{cc} \begin{array}{c} 1\\ 0 \end{array} & \begin{array}{c} 0\\ 1 \end{array} \end{array} & \begin{array}{cc} \begin{array}{c} T\\ 0 \end{array} & \begin{array}{c} 0\\ T \end{array} \end{array}\\ \begin{array}{cc} \begin{array}{c} 0\\ 0 \end{array} & \begin{array}{c} 0\\ 0 \end{array} \end{array} & \begin{array}{cc} \begin{array}{c} 1\\ 0 \end{array} & \begin{array}{c} 0\\ 1 \end{array} \end{array} \end{array}\right]$$
(6)$$F_{CT}=\left[\begin{array}{cc} \begin{array}{cc} \begin{array}{c} 1\\ 0 \end{array} & \begin{array}{c} 0\\ 1 \end{array} \end{array} & \begin{array}{cc} \begin{array}{c} \frac{\sin(wT)}{w}\\ \frac{1-\cos(wT)}{w} \end{array} & \begin{array}{c} \frac{\cos(wT)-1}{w}\\ \frac{\sin(wT)}{w} \end{array} \end{array}\\ \begin{array}{cc} \begin{array}{c} 0\\ 0 \end{array} & \begin{array}{c} 0\\ 0 \end{array} \end{array} & \begin{array}{cc} \begin{array}{c} \cos(wT)\\ \sin(wT) \end{array} & \begin{array}{c} -\sin(wT)\\ \cos(wT) \end{array} \end{array} \end{array}\right]$$

In the network model, the input consists of the measurement time series $z_{1:K}=\left\{z_{1},z_{2},...,z_{K}\right\}$, while the output is the estimated state of the maneuver target ${\tilde{s}}_{1:K}=\left\{{\tilde{s}}_{1},{\tilde{s}}_{2},...,{\tilde{s}}_{K}\right\}$, K represents the total number of time steps in the input–output time series, and w is the turning rate. To evaluate the performance of the model, we employ the mean absolute error (MAE) as a measure. We denote the Loss function with the estimated state of the maneuver target and its true state as:(7)$$Loss=\sqrt{\frac{1}{K}\sum_{k=1}^{K}\left|s_{k}-{\tilde{s}}_{k}\right|}$$

## 3. The Model for Maneuvering Target Tracking

In this section, we will discuss the deep learning algorithm applied to maneuvering target tracking based on the combined observations of azimuth and Doppler. The algorithm consists of two parts: the training phase and the testing phase. The training phase is depicted in Figure 2. Following a similar research scheme for maneuvering target tracking based on LSTM or TBN [18,19], we begin by using the observation equations and state equations to generate a large-scale trajectory dataset (LASTD) and then employ LASTD to train the neural network model. The LASTD is first followed by batch processing, and the normalized observations are input into the designed network model batch by batch. The output of the model is the predicted maneuvering target state which is converted back to its original form before. Finally, backpropagation and parameter updating of the network are performed using a loss function with the predicted states and the ground-truth states. We fine-tune the network to improve its performance in maneuvering target tracking.

### 3.1. Normalization Methods

To effectively utilize LASTD for network training and enhance the convergence speed, we normalize all generated trajectory data and their corresponding observations. In this study, we employ a hybrid normalization approach that takes into consideration the distributional characteristics of both the observation and target state data. Given that the observation data noise is assumed to be additive white noise, we adopt Gaussian normalization for the observation data. The normalization formula is expressed as follows:(8)$$z_{1:k}^{\prime }=\frac{z_{1:k}-mean\left(z_{1:k}\right)}{\mathrm{var}\left(z_{1:k}\right)}$$
where $z_{1:k}^{\prime }$, $z_{1:k}$ denotes the normalized observation segment and the corresponding generated original observation segment, respectively. $mean\left(•\right)$ denotes the mean of the segment data, and $\mathrm{var}\left(•\right)$ is the variance of the same. For the real target trajectory states, we normalize them using the min–max normalization method. The formula is denoted as:(9)$$s_{1:k}^{\prime }=\frac{s_{1:k}-\min\left(s_{1:k}\right)}{\max\left(s_{1:k}\right)-\min\left(s_{1:k}\right)}$$
where $s_{1:k}^{\prime }$, $s_{1:k}$ denote the normalized target state and the generated target state, respectively. $\min\left(•\right)$ denotes the minimum value of the data, and $\max\left(•\right)$ denotes the maximum value of the data. The partially normalized trajectory data distribution and the original trajectory data distribution are shown in Figure 3.

### 3.2. Proposed Model

Most previous studies tackling this problem of maneuvering target tracking utilized deep learning algorithms based on RNN or LSTM to model the nonlinear relationship between observation time series and target state time series. However, RNN may face challenges such as gradient explosion or vanishing during the training phase, while LSTM may not be very efficient with step-by-step prediction. Hence, the TBN was proposed for further accurate modeling with its advantages of parallel training and global processing of long sequences. Nevertheless, TBN may exhibit a weaker ability to extract local information compared to RNN and CNN [29]. 

To fully extract rich dynamic features from time series and overcome the challenges posed by the weak observation or non-observation in traditional algorithms, as well as accurately model the nonlinear relationship between observation sequences and target state sequences along with the contextual relationships among time series points in a complex motion environment, we propose a neural network called RDCINN based on one-dimensional CNN [26], as shown in Figure 4. This model architecture leverages a diverse array of convolutional filters to extract dynamic temporal features at multiple resolutions to learn the complex nonlinear relationships between temporal sequences and the contextual relationships within the temporal sequences. 

In each block module of RDCINN, the input undergoes a two-step process: splitting and interactive learning. The splitting step involves downsampling the input into two subsequences. These subsequences consist of the odd and even elements of the original sequence, respectively. This splitting operation takes advantage of the unique nature of time series, allowing the subsequences to retain most of the information present in the original sequence. The split odd and even subsequences are then individually processed by different convolutional kernels to extract valuable features and enhance the expressive power of the model. Following the splitting step, the interactive learning step compensates for any potential loss of information due to downsampling. This step involves the mutual learning of affine transform parameters between the two subsequences. The equations expressing the interactive learning step are as follows:(10)$$X_{odd}^{fst}=X_{odd}⊗\exp(conv1d(X_{even}))$$
(11)$$X_{even}^{fst}=X_{even}⊗\exp(conv1d(X_{odd}))$$
(12)$$X_{odd}^{sec}=X_{odd}^{fst}+conv1d(X_{even}^{fst})$$
(13)$$X_{even}^{sec}=X_{even}^{fst}-conv1d(X_{odd}^{fst})$$
where $X_{even}$ and $X_{odd}$ represent the subsequence of even and odd elements obtained after splitting the original sequence, respectively. conv1d represents the one-dimensional convolution layer, exp is the exponential operation applied to the sequence after convolution, and ⊗ represents the Hadamard (element-wise) product operation. $X_{odd}^{fst}$ and $X_{even}^{fst}$ represent the parity subsequences obtained after the sequence splitting, convolution, exponential, and Hadamard operations. Finally, the module generates the two parity subsequences $X_{odd}^{sec}$ and $X_{even}^{sec}$ after the interactive learning step. 

In RDCINN, an encoder–decoder architecture is employed. The encoder consists of multiple blocks organized in a binary tree structure. This structure enables the network model to have both local and global views of the entire time series, facilitating the extraction of useful temporal features. After the downsampling, convolution, and interaction operations within the encoder, the extracted features are reshaped into a new sequence representation. These reshaped features are then combined with the original time series for prediction. The decoder with a two-layer one-dimensional convolutional network layer performs the prediction based on the combined representation of the reshaped features and the original time series. This encoder–decoder architecture allows RDCINN to leverage both local and global temporal information in the input time series, enhancing its ability to capture and model complex temporal relationships for accurate prediction.

### 3.3. Test Reorganization Phase

We utilize the trained network to perform complex maneuvering target tracking on long trajectories. As illustrated in Figure 5, the process consists of the following steps: Divide the observation $z_{1:K}$ corresponding to the target states into L length segments using a sliding window approach. The window size ws is set to 16 with a moving stride st=4. This division results in $zs(n)=z_{L}$ with the length L=16 being the nth time sequence of inputs, and L can be denoted as:(14)$$L=1+\left(n-1\right)\times st:ws+(n-1)\times st$$
where *n* = 1, 2, …, (*K* − ws)/st+1, *K* is the total steps. Normalize each segment of the observation time sequence zs(n). Input the normalized observation time sequence zs(n)* into the network to obtain the predicted target state outputs. Apply denormalization on the predicted target state output ss(n)* to obtain the corresponding predicted state sequence $ss(n)=s_{L}$.

Reconstruct each segment of the predicted target state sequences after denormalization. Process the overlapping length overlap=ws−st among target state sequences ss(n) by averaging them. The reconstruction steps are expressed mathematically, as follows:(15)$${s\left(n\right)}_{1:a}=cat(s(n-1{)}_{0:b},\frac{(s(n{)}_{0:ol}+s\left(n-1{)}_{b:a}\right)}{2},{s(n)}_{ol:ws})$$
where we denote overlap as ol, $a=\left(n-1\right)\times st+ws$, b=t×(n−1), cat means that we merge the state sequence s(n−1) and s(n) together. Additionally, we set the initial states $s\left(1\right)=ss\left(1\right)$, $s\left(2\right)=ss(2)$, while $s\left(n\right)$ is the merged result of state sequences.

## 4. Simulation Experiments

In this section, we design several experimental scenarios to evaluate the superiority of our algorithm in predicting the states of strong maneuvering radar targets with the combined observations of azimuth and Doppler. Additionally, we provide a detailed explanation of the specific parameters listed in each part of the experiment.

### 4.1. Parameter Setting Details

We utilize the LASTD which consists of 450,000 trajectories with different motion models and their corresponding observations for a comprehensive evaluation of the algorithms’ performance in maneuvering target tracking tasks. The dataset was structured as follows: 150,000 samples consist of 16 s long trajectories of either uniform linear motion or uniform circular motion. Another 150,000 samples are composed of 16 s trajectories segmented into two 8 s long trajectories, every trajectory could be uniform linear motion or uniform circular motion. The remaining 150,000 samples consist of 16 s trajectories segmented into four 4 s long trajectories, while every trajectory could be either uniform linear motion or uniform circular motion. The parameters of the LASTD are listed in Table 1. We set the distance from the radar to the target about 926 m to 18,520 m, which covers the common detection range of the airport surveillance radar [30]. Aircraft rarely exceed the sound velocity in real-world scenarios, so we set the velocity of our maneuvering target in the range of −340 m/s–340 m/s [31]. According to [32], we set the turn rate w that ranges from −10°/s to 10°/s, and the standard deviation of acceleration noise is randomly sampled in the range of [8 m/s^2^, 13 m/s^2^]. The angle that ranges from −180° to 180° intersects the north and the direction from the radar to the target. The deviations of azimuth noise $\sigma_{\theta }$ and Doppler velocity noise $\sigma_{d}$ are randomly sampled in the range of [1°, 1.8°] and 1 m/s^2^ according to the funding request, respectively. Finally, we set the sample interval T at about 1 s. 

In the training process, we set the following hyperparameters for our model: the dimension E of the fully connected layer is set to 64, the binary tree height is set to 2, the convolutional layer’s kernel size, dilation rate, and group length are set to 5, 2, and 1, respectively. For the decoding layer, we have two one-dimensional convolutional layers with dimensions of 16 and 4, respectively. We use the Adam optimizer for the model training process. The weight decay rate is set to 1 × 10^−5^. The learning rate is initially set to 7 × 10^−4^, and it decays by 0.95 after each epoch. We trained 300 epochs with a batch size of 256 on a single NVIDIA 3090 GPU.

In our experiments, we compare our proposed algorithm with three existing algorithms: the LSTM network [19], the TBN model [21], and the traditional maneuvering target tracking method IMM-EKF [11]. We keep the model parameters of the LSTM network and the TBN model unchanged, as specified in their respective research papers, and train the deep learning models using the LASTD we have created.

### 4.2. Experimental Results

We first created a dataset that consists of 1500 trajectories to evaluate the performance of each baseline neural network model, as well as our model.

The dataset is similar in structure to the training set and consists of three types of trajectories with different motion patterns. Specifically, there are 500 samples of 16 s uniform linear motion trajectory or uniform circular motion trajectory, 500 samples of two 8 s uniform linear motion trajectories or uniform circular motion trajectories combined, and 500 samples of four 4 s uniform linear motion trajectories or uniform circular motion trajectories combined. The trajectory tracking performance results are shown in Table 2.

Based on the results presented in Table 2, it can be observed that our network achieves lower position mean absolute error and velocity mean absolute error results compared to the other two baseline neural networks. This demonstrates that our model, applied to the strong maneuvering target tracking domain based on the combined observations of azimuth and Doppler, outperforms the previous target tracking networks.

After that, we utilize Monte Carlo simulation to generate a 16 s strong maneuvering trajectory A. The initial state of A is [−4000 m, 4000 m, 50 m/s, −66 m/s]. This trajectory consists of four segments, each lasting 4 s and employing different motion models, which reflect sudden changes in the motion target states in real-world scenarios. The first segment of the trajectory is a 4 s uniform motion. The second segment is uniform circular motion with a turning rate w of −7°. The third segment is also a uniform circular motion but with a turning rate w of 7°. Finally, we set the last segment as a uniform motion. Additionally, we introduce azimuth observation noise as white noise with zero mean and standard deviation $\sigma_{\theta }$ of 1.8°, while the standard deviation of Doppler velocity observation noise $\sigma_{d}$ is 1 m/s. Additionally, the standard deviation of acceleration $\sigma_{s}$ is set to 10 m/s^2^. To assess the tracking performance of trajectory A, we employ our own network model as well as three other baseline algorithms. Table 3 presents the evaluation results, while Figure 6, Figure 7 and Figure 8 provide visual representations of these results.

In order to verify the applicability of our network model for tracking strong maneuvering trajectories with different step sizes, we generate trajectory B and trajectory C by conducting Monte Carlo simulations. Trajectory B is a 32 s strong maneuvering trajectory with an initial state of [−8000 m, 5000 m, −30 m/s, 21 m/s]. It consists of four segments of 8 s trajectories, each with a different model. The models for each segment are as follows: uniform circular motion with a turning rate w of 6°, uniform motion, uniform circular motion with a turning rate w of −5°, and uniform motion, respectively. Trajectory C is a 64 s strong maneuvering trajectory with an initial state of [−5000 m, 5000 m, 30 m/s, −23 m/s]. It also consists of four segments of 16 s trajectories with different motion models. The motion models for each 16 s trajectory are as follows: uniform circular motion with a turning rate w of −1°, uniform motion, uniform circular motion with a turning rate w of 2°, and uniform motion, respectively. Keeping the standard deviation setup as how trajectory A was set up as the same, we then evaluate the tracking performance of trajectories B and C using our network model and three baseline algorithms. The evaluation results are presented in Table 4 and Table 5. Additionally, Figure 9, Figure 10, Figure 11, Figure 12, Figure 13 and Figure 14 provide visual representations of these results.

In our simulation experiment, we find that the change of the observation information of the noisy azimuth in the highlighted time fragment marked in the figure is extremely subtle, when the trajectory is in the CV motion state; it results in the change of the observation information of the associated Doppler velocity being also subtle and the target is in the non-observation state. Then, the target is in the weak observation state when in the CT motion state, where there is only observation information of the Doppler velocity playing a role in the tracking scenario. The highlighted place in Figure 6 is the target unobservable state, and the highlighted place in Figure 9 and Figure 12 is the weak observation state of the target.

We then experimented with different azimuth noise standard deviation values on trajectory C to further test the generalization ability of our model algorithm, as shown in Table 6. When we use the values within the standard deviation range of azimuth noise set by LASTD for testing, we find that the position MAE and velocity MAE of the target can still be kept at a small value in this case, indicating that the performance of the model algorithm can still have the desired effect. When the noise standard deviation of the azimuth angle is adjusted to 2.8, the position MAE and velocity MAE of the target increase greatly, and when adjusted to 3.8, the position MAE and velocity MAE of the target can achieve our expected effect, which shows that our model algorithm still has a certain generalization ability.

The experimental results demonstrate that our model achieves superior trajectory tracking performance compared to other algorithms. This is particularly noticeable when tracking strong maneuvering targets under the combined observations of azimuth and Doppler.

## 5. Conclusions

In this paper, we propose a deep learning algorithm called RDCINN for tracking strong maneuvering targets with the combined observations of azimuth and Doppler. We utilize LASTD generated by the motion models to train RDCINN to learn the nonlinear mapping relationship between observations and target states and facilitate accurate offline estimation of target states in complex maneuvering scenarios despite noisy observations. Simulation results demonstrate that our algorithm not only addresses the limitations of traditional target tracking algorithms, which struggle to update target states due to weak observation or non-observation, but also outperforms two previous deep learning algorithms applied to maneuvering target tracking. It is important to note that our algorithm is currently limited to two-dimensional target tracking scenarios, and future work will focus on extending its application to three-dimensional scenarios. Furthermore, there is relatively limited research on the utilization of temporal convolutional networks in the field of target tracking. Future work will involve gaining a deeper understanding of temporal convolutional networks and further improving state estimation accuracy.

## Figures and Tables

**Figure 1 sensors-24-00263-f001:**
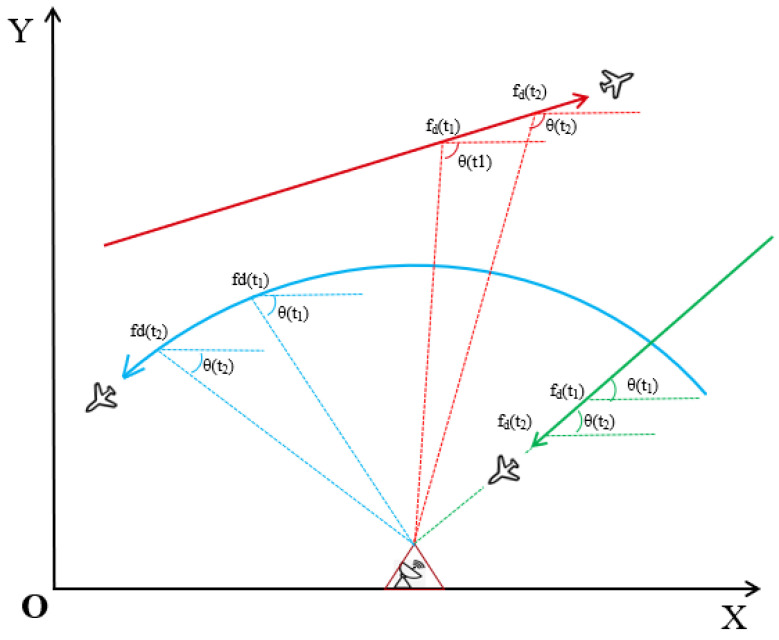
The relative positions of the observation point and the maneuvering target in the two-dimensional X-Y plane. The green trajectory illustrates the movement of a maneuvering target at a uniform or uniformly variable speed along the target and the observation point. The blue trajectory demonstrates the trajectory of a target moving circularly at a uniform velocity around the observation point. The red trajectory shows the movement of a maneuvering target relative to the observation point.

**Figure 2 sensors-24-00263-f002:**
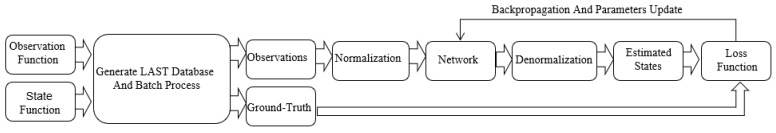
Flowchart of the algorithm training process. Generate target state and observation pairs for network training with motion and observation models first, then train with batch-normalized observation as inputs, and update the parameters with backpropagation of the loss error between the predicted state and the true state iteratively.

**Figure 3 sensors-24-00263-f003:**
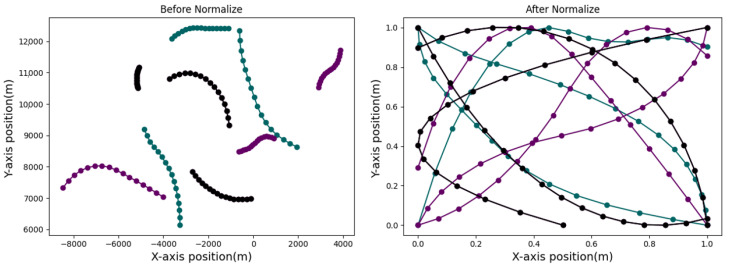
Min–max normalization of trajectory segments. The (**left side**) of the figure shows the original trajectories generated by 9 different motion models. The black trajectory line represents only one maneuver, the purple trajectory has two maneuvers, and the green trajectory has four maneuvers. On the (**right side**), the distributions of the normalized trajectory data can be observed. The X and Y ranges of the trajectories are restricted to the interval [0, 1], which ensures that the LASTD does not have excessively large ranges to pose difficulties during network training.

**Figure 4 sensors-24-00263-f004:**
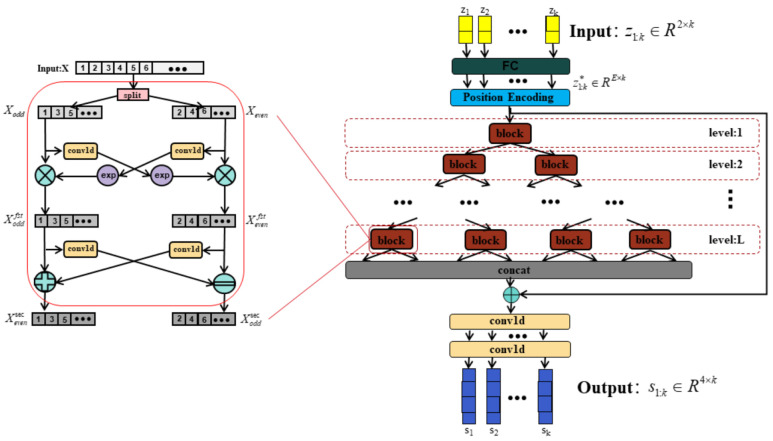
Model architecture of RDCINN. On the left is a diagram of a sub-block structure of the binary tree network architecture, which performs splitting, convolution, and interactive learning operations on the input sequence X sequentially and finally outputs the odd and even subsequences $X_{odd}^{sec}$ and $X_{even}^{sec}$. The input to the binary tree network is the 2-dimensional sequence of normalized observations *z*_1:k_. It is first mapped through the fully connected layer into the E-dimension and the final outputs obtained through the network as a 4-dimensional sequence of predicted target states *s*_1:k_.

**Figure 5 sensors-24-00263-f005:**
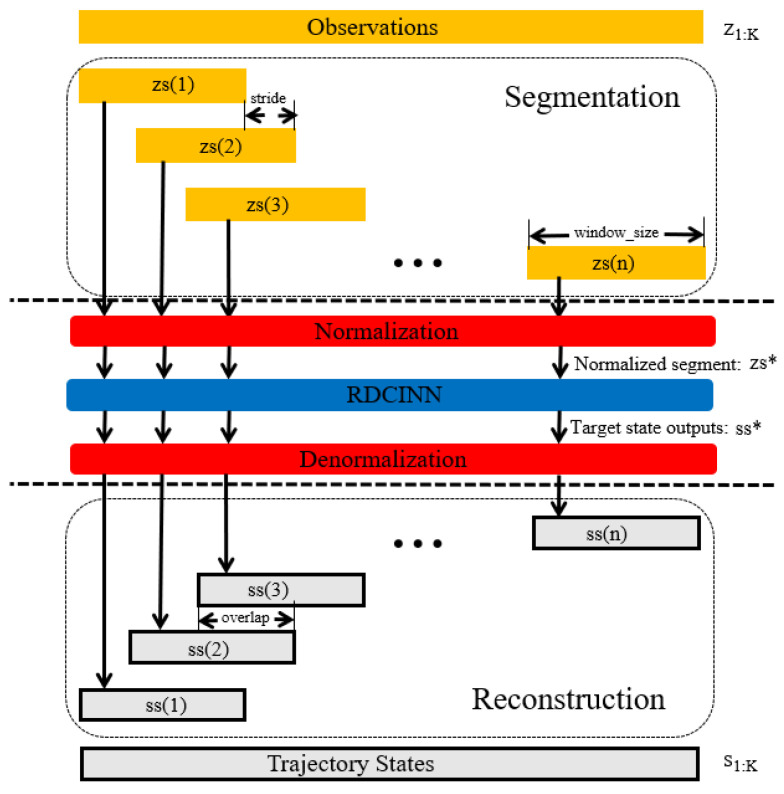
Trajectory segmentation and reconstruction consist of the following steps: Divide the observations $z_{1:K}$ corresponding to the target states using a sliding window approach. Normalize each segment of the observation time sequence zs(n) as zs*(n). Apply denormalization on the predicted target state outputs ss(n)* to obtain the corresponding predicted state sequence ss(n). Finally, reconstruct each segment of predicted target state sequences after denormalization.

**Figure 6 sensors-24-00263-f006:**
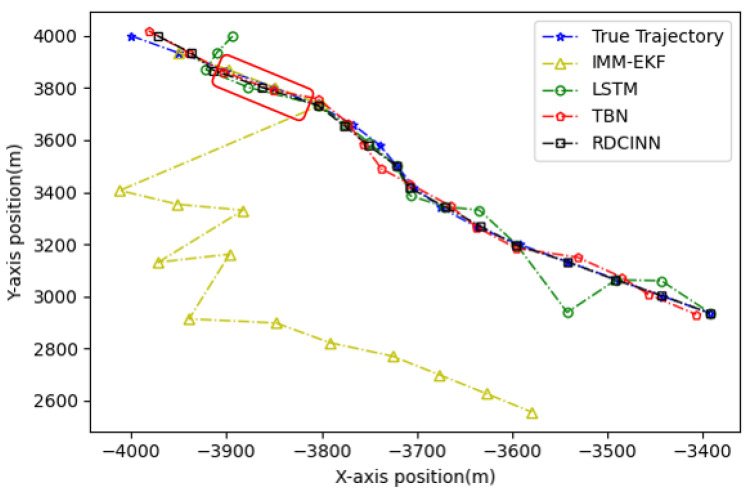
Tracking trajectory results of the trajectory A using different algorithms on the X-Y plane.

**Figure 7 sensors-24-00263-f007:**
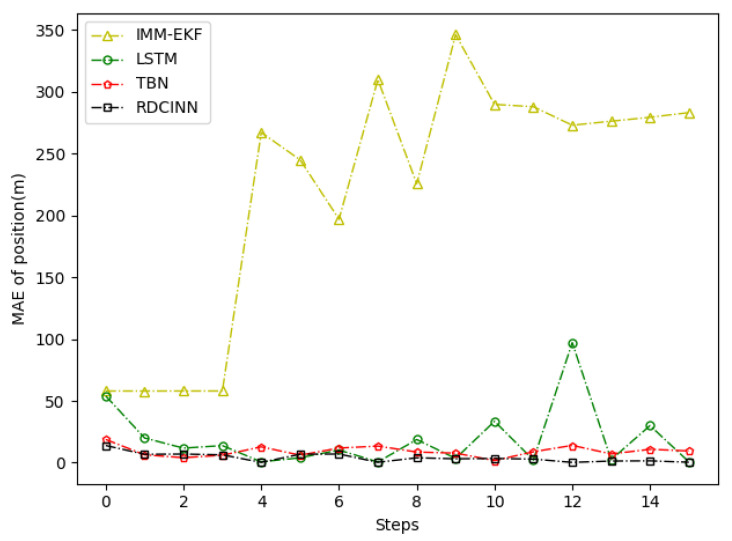
MAE results of position tracking by different algorithms for trajectory A.

**Figure 8 sensors-24-00263-f008:**
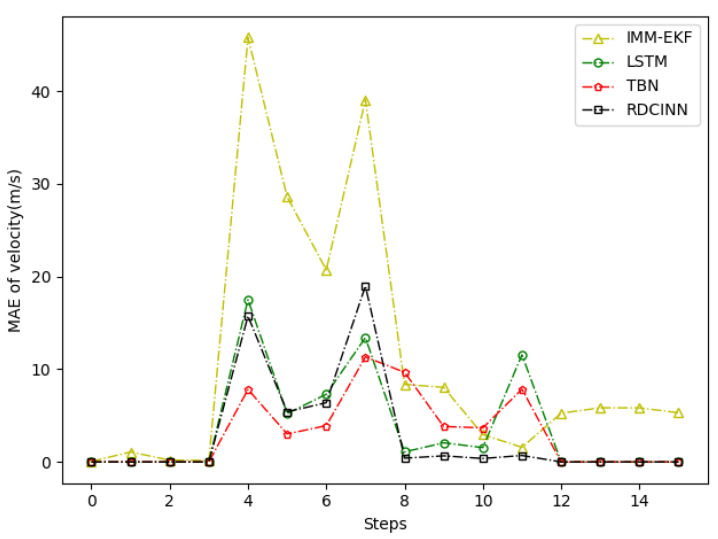
MAE results of velocity tracking by different algorithms for trajectory A.

**Figure 9 sensors-24-00263-f009:**
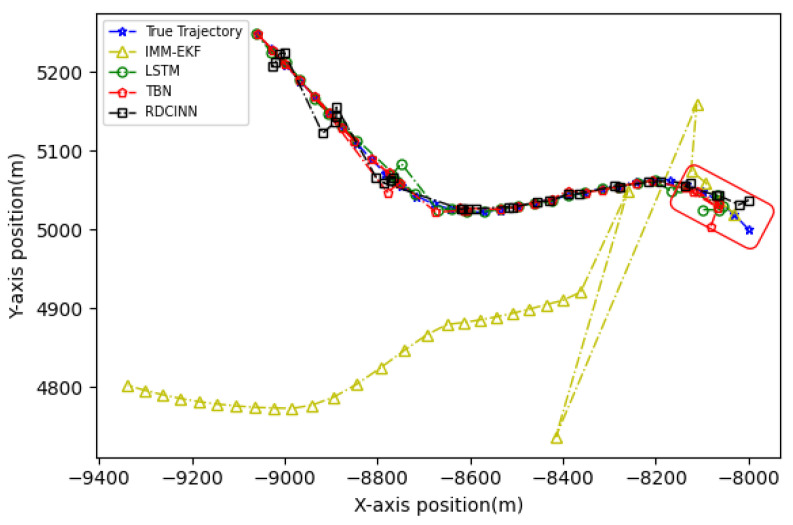
Tracking trajectory results of the trajectory B using different algorithms on the X-Y plane.

**Figure 10 sensors-24-00263-f010:**
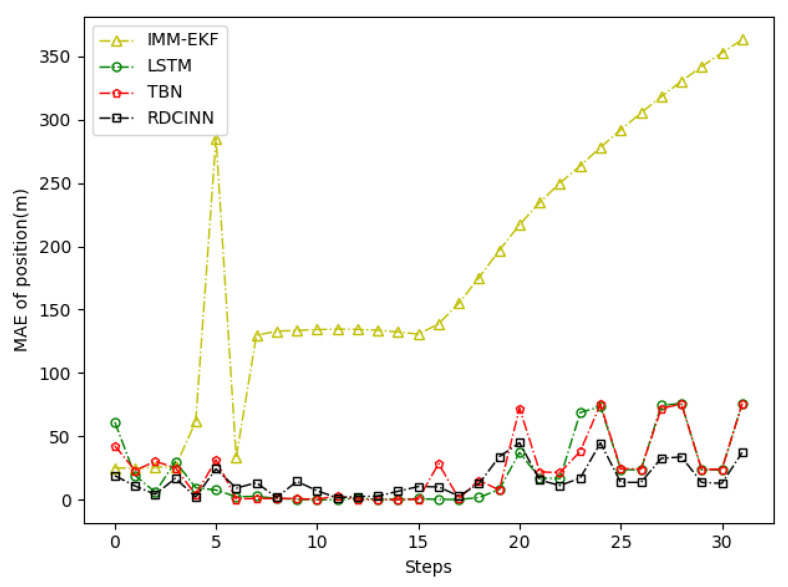
MAE results of position tracking by different algorithms for trajectory B.

**Figure 11 sensors-24-00263-f011:**
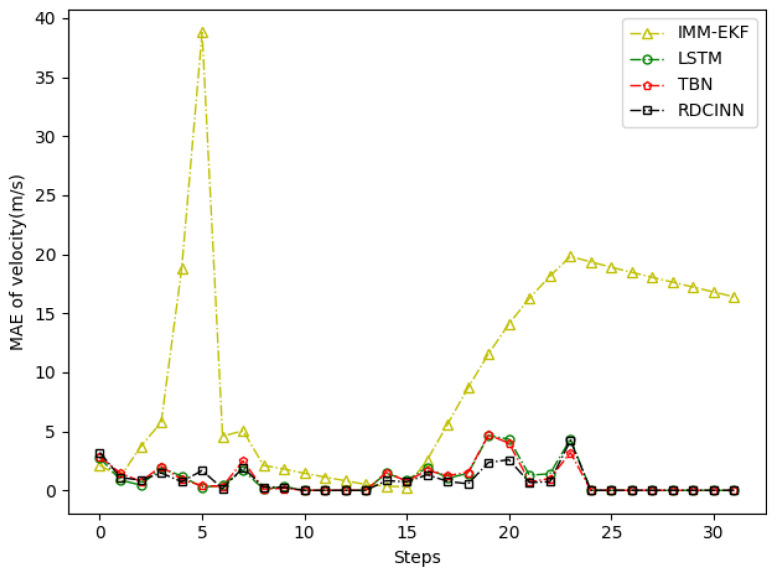
MAE results of velocity tracking by different algorithms for trajectory B.

**Figure 12 sensors-24-00263-f012:**
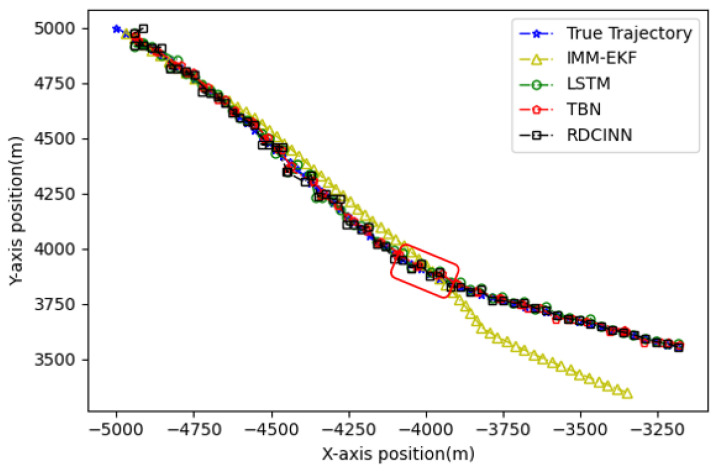
Tracking trajectory results of the trajectory C using different algorithms on the X-Y plane.

**Figure 13 sensors-24-00263-f013:**
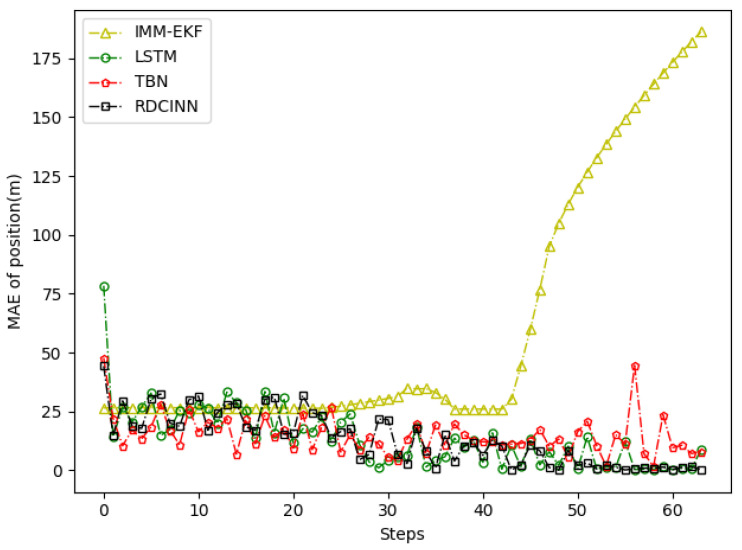
MAE results of position tracking by different algorithms for trajectory C.

**Figure 14 sensors-24-00263-f014:**
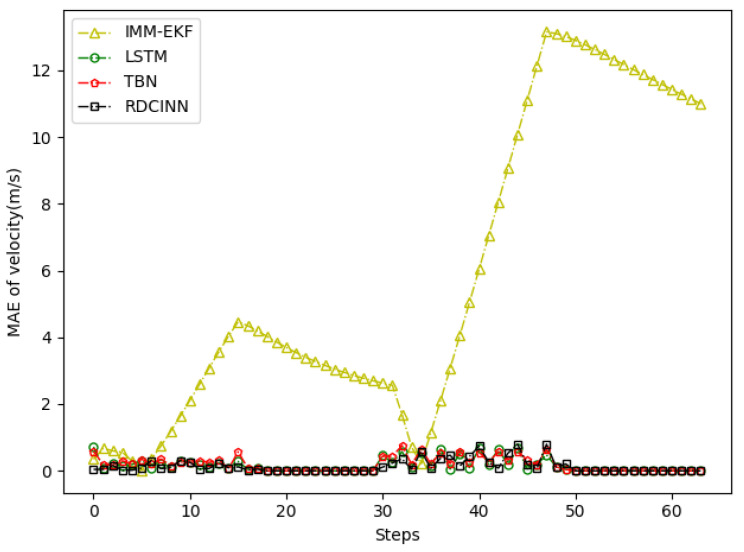
MAE results of velocity tracking by different algorithms for trajectory C.

**Table 1 sensors-24-00263-t001:** Parameters of LASTD.

Parameters	Value
Distance Range	[926 m, 18,520 m]
Angle Range	[−180°, 180°]
Velocity Range	[−340 m/s, 340 m/s]
Turn Rate (w)	[−10°/s, 10°/s]
The Standard Deviation of Acceleration Noise ($\sigma_{s}$)	[8 m/s^2^, 13 m/s^2^]
The Standard Deviation of Azimuth Noise ($\sigma_{\theta }$)	[1°, 1.8°]
The Standard Deviation of Doppler Noise ($\sigma_{d}$)	1 m/s
Sampling Time Interval (T)	1 s

**Table 2 sensors-24-00263-t002:** Tracking performance results of several neural network algorithms for trajectory segments.

	MAE of Position (m)	MAE of Velocity (m)	RMSE of Position (m)	RMSE of Velocity (m)
LSTM	58.73	8.84	137.68	18.12
TBN	44.16	6.82	120.53	15.60
RDCINN	42.76	6.35	119.82	16.03

**Table 3 sensors-24-00263-t003:** Tracking performance of several target tracking algorithms on trajectory A.

	MAE of Position (m)	MAE of Velocity (m)	RMSE of Position (m)	RMSE of Velocity (m)
IMM + EKF	219.53	11.17	249.64	18.06
LSTM	18.92	3.73	43.58	7.72
TBN	9.21	3.19	11.09	5.51
RDCINN	4.07	3.04	7.13	7.45

**Table 4 sensors-24-00263-t004:** Tracking performance of several target tracking algorithms on trajectory B.

	MAE of Position (m)	MAE of Velocity (m)	RMSE of Position (m)	RMSE of Velocity (m)
IMM + EKF	184.08	10.25	215.83	14.19
LSTM	21.56	1.02	35.66	2.22
TBN	24.01	0.99	36.78	2.15
RDCINN	15.66	0.82	22.43	1.53

**Table 5 sensors-24-00263-t005:** Tracking performance of several target tracking algorithms on trajectory C.

	MAE of Position (m)	MAE of Velocity (m)	RMSE of Position (m)	RMSE of Velocity (m)
IMM + EKF	60.82	5.73	83.92	7.33
LSTM	13.65	0.16	20.13	0.35
TBN	15.02	0.19	20.32	0.30
RDCINN	13.33	0.13	18.48	0.32

**Table 6 sensors-24-00263-t006:** Tracking performance of several standard deviations $\sigma_{\theta }$ on trajectory C.

The Value of Standard Deviation $\sigma_{\theta }$ (degree)	MAE of Position (m)	MAE of Velocity (m)
$\sigma_{\theta }$ = 0.8	14.67	0.38
$\sigma_{\theta }$ = 1.2	16.6	0.21
$\sigma_{\theta }$ = 2.8	31.51	0.39
$\sigma_{\theta }$ = 3.8	18.26	0.35

## Data Availability

Data are contained within the article.

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
