# Peer review of "Time Convolutional Network-Based Maneuvering Target Tracking with Azimuth–Doppler Measurement"

_sensors, 2024, doi:10.3390/s24010263_

Round 1

Reviewer 1 Report

Comments and Suggestions for Authors

The paper studies the problem of maneuvering target tracking, and the topic is meaningful. The paper proposes to use an RDCINN method to improve tracking accuracy, 

It has some reference value. Comments as follows:

1. The discussion on the mathematical model of the tracking problem in the paper is too simplistic and requires more details. And directly express the velocity vector in formula (1) as

Doppler information is not very accurate. 

2. It is necessary to explain the parameter settings in Table 1, and it is best to provide an actual scenario with the parameters in the table.

3. When analyzing and comparing the simulation results of several methods in the paper, it would be more convincing to compare them with similar results in other literature.

Reviewer 2 Report

Comments and Suggestions for Authors

Reviewing article “Time Convolutional Network Based Maneuvering Target Tracking with Azimuth-Doppler Measurement

Sensors- 2755454

Decision: The authors propose a deep learning algorithm called RDCINN for target tracking with Azimuth-Doppler measurement, they claim that the method overcomes the challenges of weak observations or no-observation scenarios. The proposed method sounds interesting. However, there are a few comments I would like the authors to address in a revised version and I will recommend it to be published in Sensors after some improvements are made to the discussion and presentation of the paper. Please find below the comments.

Comments:

  1. In the literature review, there are other filters used for tracking purposes and they demonstrated capabilities to tackle some challenges that are addressed in this manuscript, it will be useful to add them in the benchmarking analysis or at least consider them in a brief discussion. Such as:

Shadowing filter

 [1] “Tracking Rigid Bodies Using Only Position Data: A Shadowing Filter Approach Based on Newtonian Dynamics”. Digit. Signal Process. 201767, 81–90.”

Shadowing filter, as shown in [1] , has been tested and demonstrated capability to tackle weak observation or no-observation scenarios as the core advantage of the proposed method, this nominates the shadowing filter as a potential approach to be used for the purpose of the authors’ application.

Particle filter

[2] “Particle Filters for Positioning, Navigation and Tracking”; Linköping University Electronic Press: Linköping, Sweden, 2001.”  

[3] “Analysis of Selection Methods for Cost-reference Particle Filtering with Applications to Maneuvering Target Tracking and Dynamic Optimization”. Digit. Signal Process. 200717, 787–807” 

2.     The proposed method is based on training the deep learning model, and this requires the generation of a large-scale trajectory data set (LASTD). My concern here, is this a limitation for real-world applications. Will the LASTD data set cover all possible maneuvering scenarios in real practice? I would suggest the author provide a discussion on this point and address clearly if this is a limitation of the proposed method or any deep learning tracking method.

3.     In equation (4), \eta_{s,k}, are the noises in the two coordinates the same? Could you please clarify? 

4.     In line 153, there is a typo “andwis”

5.     Is there a reference for the odd-even subsequencing?

6.     In line 293, what do the authors mean by “different motion laws”? do they mean different motion models?

7.     In line 320, “The data set is similar in structure to the training set”, what if the data set has a different structure? How is this applicable in real-world practice? Could the author provide a clarification? This is also related to point 2 above.

8.     In the Experimental Results, how does the value of the standard deviation affect the performance? Has the method been tested for different/larger noises? At least, could you please provide a discussion of why these values have been chosen?

9.     In the Experimental Results and Figures 6 to 14, could the authors highlight where the weak observations or no-observation instances occur?

Round 2

Reviewer 1 Report

Comments and Suggestions for Authors

The author's response to Comment 3 is not very good. I intended to want the author to compare what others have achieved, not to compare the three ways you have implemented yourself.

Reviewer 2 Report

Comments and Suggestions for Authors

The authors have addressed the comments and I recommend the manuscript to be published in Sensors.

Well done and congratulations!

Author Response

Thank you very much for your suggestions and appreciation!